# The Global Research Status and Trends in Ice and Snow Sports Injuries from 1995 to 2022: A Bibliometric and Visualized Analysis

**DOI:** 10.3390/ijerph20042880

**Published:** 2023-02-07

**Authors:** Wupeng Zhang, Hua Li, Daofeng Wang, Gaoxiang Xu, Cheng Xu, Jiantao Li, Licheng Zhang, Peifu Tang

**Affiliations:** 1School of Medicine, Nankai University, No. 94 Weijin Road, Tianjin 300071, China; 2Senior Department of Orthopedics, The Fourth Medical Center of Chinese PLA General Hospital, Beijing 100048, China; 3National Clinical Research Center for Orthopedics, Sports Medicine and Rehabilitation, Beijing 100853, China

**Keywords:** ice and snow sports, winter sports, sports injuries, bibliometrics, visualized analysis

## Abstract

Objective: The aim of the present study is to focus on the most popular winter sports programs, and to investigate the global research status and trends in sports-related injuries. Methods: The Web of Science (WoS) Core Collection database was chosen as original data and used for extracting publications on ice and snow sports injuries on 18 February 2022. Articles published in English between 1995 to 2022 were selected for this study. Results: Finally, for the topic search, a sum of 1605 articles were extracted and used for further analysis. The country and journal ranked first—in terms of total number, total citations and the highest H-index of publications—were the USA and American Journal of Sports Medicine, respectively. The affiliation with the most cited publications was the Norwegian School of Sport Sciences. The most influential first author with the most citations (2537 times), the greatest average citations per article (65.05 times) and the highest H-index (26) was Bahr R. Articles were divided into five main clusters based on keyword analysis: injuries study, head and neck damage study, risk study, therapy study and epidemiology study. Studies related to epidemiology and brain damage in ice and snow sports will continue to be research hot topics. Conclusions: In conclusion, our study indicates that the ice and snow sports injuries research domain is more prevalent in North America and Europe. This study contributes to a comprehensive understanding of ice and snow sports injuries and provides hotspot directions.

## 1. Introduction

The Winter Olympics promote the leapfrog development of global ice and snow sports and the ice and snow industry, enriching the content of fitness sports and improving people’s health. With the promotion and popularization of the Winter Olympic Games, ice and snow sports are becoming popular worldwide. However, because of its characteristics of high speed and difficult motions, practicing ice and snow sports is more likely to result in sports-related injuries [1,2]. The increase in the number of people participating in snow and ice sports will inevitably raise the incidence of sports injuries. Injuries associated with acute trauma in ice and snow sports primarily involve the head, spine and upper and lower extremities, and seriously affect the health and career of athletes [3,4,5]. In order to detect and count the sports injuries that occur in participating athletes and to further protect the health of athletes, the International Olympics Committee (IOC), in cooperation with National Olympic Committees (NOC) and International Sports Federations, initiated and developed the Olympic injury and illness surveillance system in 2008 [6,7]. However, at the same time, what needs to be monitored, the main current international concerns and how to monitor scientifically still need to be researched and explored. Therefore, the present study aims to perform a bibliometric analysis of ice and snow sports injuries research.

A bibliometric study is used for the analysis of co-authorship, co-occurrence, bibliographic coupling and co-citation of previous publications by applying a range of mathematical and statistical methods to quantitatively and qualitatively analyze and measure articles and other forms of publications. Bibliometric studies are widely used in various medical research fields [8,9,10,11,12,13].

However, as far as we know, there have been no studies related to bibliometric and visualized analysis of winter sports-related injuries up until now. Hence, the aims of this study are to investigate the global research status and trends of winter sports through bibliometric methods, to gain a more comprehensive insight into the changes in research trends in different research hotspots (incidence, characteristics, mechanisms and risk factors, etc.) of winter sports injuries in recent decades, in order to scientifically understand and further mediate the risks of injury and improve safety, and to provide directional guidance and research ideas for further research on winter sports injuries.

## 2. Methods

### 2.1. Data Source and Search Protocol

The publicly available data of publications were extracted through a search in the Web of Science (WoS) Core Collection, which is considered the optimum database for conducting bibliometric quantitative evaluation analysis [14]. The publications included in this study were published from 1995 to 2022. To ensure the quantity and quality of the articles, we selected a number of sports that were most relevant to specific injuries and used topics to search on 18 February 2022. The publications data were determined by two researchers. The search protocols and terms were as follows: TS = (ice sports injur* OR snow sports injur* OR winter sports injur* OR skiing injur* OR skating injur* OR ice hockey injur* OR snowboard injur* OR sled injur*). The type of document was refined to articles. The top three WoS research orientations that were closely related to this study were selected: sport sciences, orthopedics and surgery. Articles published in English were selected for this study. The extracted information included: publication years, authors, affiliations, publication titles, countries/regions, number of articles, total citations, average citations per item, and H-index. The flow chart for article inclusion and exclusion is shown in the Appendix A.

### 2.2. Bibliometric and Visualized Analysis Methods

The publication years, total citations, average citations per item, H-index and other basic characteristics were obtained by using the intrinsic analysis function of WoS. The H-index was proposed by the American theoretical physicist Jorge E. Hirsch in 2005—defined as whether h of his or her Np papers have at least h citations each and the other (Np–h) papers have ≤h citations each—as a useful and valuable index to provide a quantitative assessment of the importance, significance and broad impact of a scientist’s cumulative research contributions and outputs [15,16,17]. Compared with the total impact factor and total citations, the H-index aims to better describe the scientific productivity and influence of authors, affiliations, countries/regions or journals [16,17].

VOSviewer software 1.6.18 (Leiden University, Leiden, The Netherlands) was used for constructing and visualizing bibliometric networks based on co-authorship, co-occurrence, bibliographic coupling and co-citation relations [18,19,20]. VOSviewer provides a better understanding and easier interpretation of different terms and clusters through the construction of network visualization maps, overlay visualization maps and density visualization maps. Overlay visualizations can, for instance, be used to show developments over time. Density visualizations provide a quick overview of the main areas in a bibliometric network. The publications data extracted from WoS were imported into VOSviewer to analyze the number of articles, understand the prominent authors, affiliations, journals and countries/regions, mine the highly influential articles, understand the research frontiers and hotspots and explore the future research direction in this domain. The size of the nodes and the thickness of the connecting lines correlate with the importance of the type and unit of the analysis. Associations between authors, affiliations and countries/regions are visualized by weighted total link strength (TLS) lines.

## 3. Results

### 3.1. Publication Output and Growth Trends

As of 18 February 2022, for the topic search, a sum of 1605 articles were extracted from WoS. Figure 1 shows the top fifteen countries/regions, affiliations, journals in number of articles, and the growth trends of publications over the past 28 years.

The United States was at the forefront of ice and snow sports injuries research, accounting for 41.2% of total publications (661), with a much larger number of papers than other countries and regions. After the United States, Canada ranked second in number of publications (27), accounting for 17.3% of total publications, followed by Germany (128, 8.0%) (Figure 1a). Figure 1f shows the global distribution of the number of articles. University of Calgary ranked first in number of articles (81), followed by University of Innsbruck (68) and Norwegian School of Sport Sciences (67) (Figure 1b). The top 15 affiliations with the most articles were all located in North America and Europe. The journal with the highest number of articles was American Journal of Sports Medicine (147), followed by British Journal of Sports Medicine with 138 articles and Clinical Journal of Sport Medicine with 103 articles on the research of winter sports injuries. The top 15 journals with the most publications are shown in Figure 1c. The author with the highest number of articles was Ruedl G (43), followed by Burtscher M with 40 articles and Bahr R with 39 articles (Figure 1d). Since 2010, the number of articles published in this field has been on an overall upward trend, peaking at 113 articles in 2021 (Figure 1e). Figure 1g shows that the linear regression-predicted growth model equation based on historical data was Y = 2.649 × X − 5258, with X and Y representing the year and the predicted number of publications, which could help predict future trends. The number of publications is expected to reach 119 in 2030. A total of 1605 articles were written and published in English.

### 3.2. Publication Quality of Authors, Affiliations, Countries/Regions and Journals

The total citations, average citations per item and H-index of the top 15 countries/regions, affiliations, authors and journals with the most articles in the field of ice and snow sports injuries are illustrated in Table 1, Table 2, Table 3 and Table 4 and Figure 2a–d. The United States ranked first with the highest total citations (20,204), and articles with the highest average citations per item were from Norway (43.43). The United States ranked first in H-index with 72, followed by Canada (44) and Norway (34). Articles from Norwegian School of Sport Sciences had the highest total citations (3306) and H-index, which was 32. The Pennsylvania Commonwealth System of Higher Education ranked first in average citations per item (51.53). The most influential first author with the most citations (2537), the greatest average citations per article (65.05) and the highest H-index (26) in this domain was Bahr R (Norwegian School of Sport Sciences). Articles from American Journal of Sports Medicine had the highest total number of citations (8584), the greatest average citations per article (58.39) and the highest H-index (52), followed by British Journal of Sports Medicine with 5728 times, 41.51 times and 40, respectively.

### 3.3. Co-Authorship Analysis

Co-authorship analysis was used to analyze authors’ collaboration based on co-authored articles. Figure 3a showed a total of 837 authors were identified with a minimum of two articles. The author with the greatest TLS was Emery CA (126), followed by Bahr R with 97, Ruedl G with 96, Engebretsen L with 91 and Meeuwisse WH with 88. Figure 3b shows that a total of 525 affiliations were identified with a minimum of two articles. The affiliations with the greatest TLS over 100 were the University of Calgary (219), University of North Carolina (127), Harvard Medical School (113) and Norwegian School of Sport Sciences (108). Figure 3c shows that a total of 46 countries/regions were identified with a minimum of two articles. The top five countries/regions with the greatest TLS were as follows: Canada (212), the United States (194), Switzerland (134), England (120) and Germany (110).

### 3.4. Co-Occurrence Analysis

Co-occurrence analysis was used to reflect the strength of association between key words, and to determine the research hotspots, composition and paradigms of the disciplines or fields represented by these words, and to analyze the development process and structural evolution of the subject areas horizontally and vertically [21]. Figure 4a shows that a total of 233 identified keywords with a minimum of five occurrences were divided into five main clusters: injuries study (red color), head and neck damage study (blue color), risk study (green color), therapy study (yellow color) and epidemiology study (purple color). These categories are based on the keyword clustering algorithm generated within VOSviewer. The same color indicates that the articles are similar in terms of research content. The red cluster (123 items) includes keywords such as injuries, injury, biomechanics and performance, etc. The blue cluster (100 items) includes keywords such as ice hockey, concussion and sports, etc. In the green cluster (98 items), the most frequently appearing keywords are risk, skiing, children and snowboarding. In the yellow cluster (91 items), the most frequently appearing keywords are anterior cruciate ligament, knee and management. In the purple cluster (77 items), the most frequently appeareing keywords are epidemiology, prevention and players. An overlay visualization map was used to show the developments of keywords over time (Figure 4b). The overall trend gradually transited from injury and risk studies to therapy, epidemiology and brain damage studies over time.

### 3.5. Bibliographic Coupling Analysis

Bibliographic coupling analysis was conducted to reflect the relationship and similarity between two papers that cited a common third article [22,23]. Figure 5a shows that a total of 140 affiliations were identified with a minimum of five articles. The affiliations with the greatest TLS were the University of Calgary (56,129), Norwegian School of Sport Sciences (33,832), University of North Carolina (30,354), Mayo Clinic (23,939) and International Olympic Committee (21,417). Figure 5b shows that a total of 128 authors were identified with a minimum of five articles. The author with the greatest TLS was Engebretsen L (22,861), followed by Bahr R with 22,095, Steffen K with 96, Meeuwisse WH with 18,178 and Emery CA with 17,152. Figure 5c shows that a total of 65 journals were identified with a minimum of five articles. The journal with the greatest TLS was British Journal of Sports Medicine (25,936), followed by American Journal of Sports Medicine with 23,472, Clinical Journal of Sport Medicine with 16,642, Scandinavian Journal of Medicine Science in Sports with 8764 and Journal of Athletic Training with 8422. Figure 5d shows that a total of 32 countries/regions were identified with a minimum of five articles. The top five countries/regions with the greatest TLS were as follows: the United States (93,068), Canada (76,781), Norway (49,342), Switzerland (45,642) and Austria (27,947).

### 3.6. Co-Citation Analysis

Co-citation analysis was defined as the frequency with which two documents were cited together by other documents [24]. Figure 6a shows that a total of 256 journals were identified with a number of citations at least 20 times. The journal with the greatest TLS was American Journal of Sports Medicine (112,796), followed by British Journal of Sports Medicine with 83,647, Clinical Journal of Sport Medicine with 42,745, Journal of Athletic Training with 30,782 and Medicine and Science in Sports and Exercise with 29,218. Figure 6b shows that a total of 260 authors were identified with a number of citations at least 20 times. The top five authors with the greatest TLS were as follows: Mccrory P (3722), Ruedl G (3592), Florenes TW (3366), Emery CA (3229) and Junge A (3061).

## 4. Discussion

We identified, through topic research, 1605 ice and snow sports injuries articles that were published in all journals in the WoS from 1995 and 2022. This bibliometric analysis of global research trends in the field of ice and snow sports injuries showed a steady increase in the number of publications over the past 28 years. These articles also covered a wide range of topics that appeal to all professionals. Through these articles, we can provide the necessary summary and overview of the changes in research trends in different research hotspots (incidence, characteristics, mechanisms and risk factors, etc.) of high-level events and winter sports injuries in recent decades.

Bibliometrics was introduced by Paul Otlet and has been popular among researchers in supporting quantitative analysis for understanding the literature and global research status and trends in a domain of interest [25,26]. In orthopedics, scientific methodology and approaches are important in fighting against occurrence and taking control over injury interventions. Within the aim of the study, a bibliometric and visualized analysis was conducted regarding ice and snow sports injuries.

Our study presented an objective and comprehensive overview of the trends and development in ice and snow sports injuries from 1995 to 2022. The annual number of articles related to ice and snow sports injuries has been increasing gradually over the past 28 years and will continue to increase in the next decade. It is speculated that the annual number of articles in this domain will continue to increase to 119 by 2030. The overlay visualization map is color-coded based on the year in which the keywords appear in the articles, and shows developments over time. From the explosion of keywords, we speculate that more and more researchers will focus on studies related to the epidemiology, brain injury and therapy regarding snow and ice sports injuries. Epidemiology-related studies can guide athletes and participants to be alert and avoid various potential injury factors during sports. Brain injuries have been attracting attention because of their greater potential for life-threatening injuries compared to other parts of the body. Therapy-related studies allow for the timely diagnosis and treatment of ice and snow sports injuries in combination with program characteristics and movement mechanisms in order to achieve optimal rehabilitation. These categories are scattered but each has its own focus. This prompts us to pay systematic attention to snow and ice sports injuries. For training guidance, it is important to focus on scientific protection and a multidisciplinary and integrated response plan.

At present, countries/regions, affiliations, authors and journals that mainly focus on snow and ice sports injury research are highest in number in developed Western countries. This is attributed to the long duration, wide audience and high level of winter sports programs in developed Western countries. According to bibliometric analysis, the Bahr R team, the Engebretsen L team, the Meeuwisse WH and Emery CA team, the Ruedl G and Burtscher M team and the Kerr ZY team still lead in this domain. University of Calgary, Norwegian School of Sport Sciences, University of North Carolina and University of Innsbruck still dominate the current direction of ice and snow sports injuries. American Journal of Sports Medicine, British Journal of Sports Medicine and Clinical Journal of Sport Medicine remain in an unassailable position for ice and snow sports injuries. The analysis results indicate that the journals and authors mentioned above are more deserving of attention. The above teams’ studies in the field of ice and snow sports injuries are more likely to reflect the latest progress, and the breakthrough results and findings in this field are more likely to be reported in the above journals. The publication analysis shows that communication and cooperation between various international teams are increasing, and more exchanges and cooperation are needed to promote research in this area in the future.

Articles on ice and snow sports injuries in this study were extracted from the WoS Core Collection database. The bibliometric and visualized analyses are more objective and comprehensive compared to other studies and could help to understand research hotspots and explore future research directions. However, there are some limitations inherent in this study that must be considered. This study is limited to articles published after 1995, and only the top three WoS research orientations were selected in the study and analyzed. Therefore, partially important articles on ice and snow sports injuries were excluded. Additionally, we only selected a few sports that are most relevant to relevant injuries, so the accuracy of the analysis results is partially affected. Third, the bibliometric analysis is limited by database variation, i.e., PubMed, Scopus and Google Scholar are not included in our study [27]. Future research should include a more comprehensive orientation and a broader range of ice and snow sports. Finally, a phenomenon that also affects our findings is that some influential articles are cited a limited number of times [28].

## 5. Conclusions

This study analyzes publications regarding ice and snow sports injuries around the world by using bibliometric methods to reveal the current global research status and related trends. Since 2010, study on ice and snow sports injuries has been experiencing steady growth, mainly conducted in North America and Europe. The top three productive journals in the field of ice and snow sports injuries are American Journal of Sports Medicine, British Journal of Sports Medicine and Clinical Journal of Sport Medicine. Studies related to epidemiology, brain damage and therapy in ice and snow sports will continue to be research hot topics. Particularly, bibliometric and visualized studies focusing on ice and snow sports injuries can be a valuable tool for understanding the research frontiers and hotspots and exploring future research directions in this domain.

This study provides insight into how to enhance the awareness level of athletes, coaches and medical workers on ice and snow sports injuries, so as to scientifically and reasonably arrange training and skill actions to avoid injury risks and be alert to various potential injury factors during competition.

## Figures and Tables

**Figure 1 ijerph-20-02880-f001:**
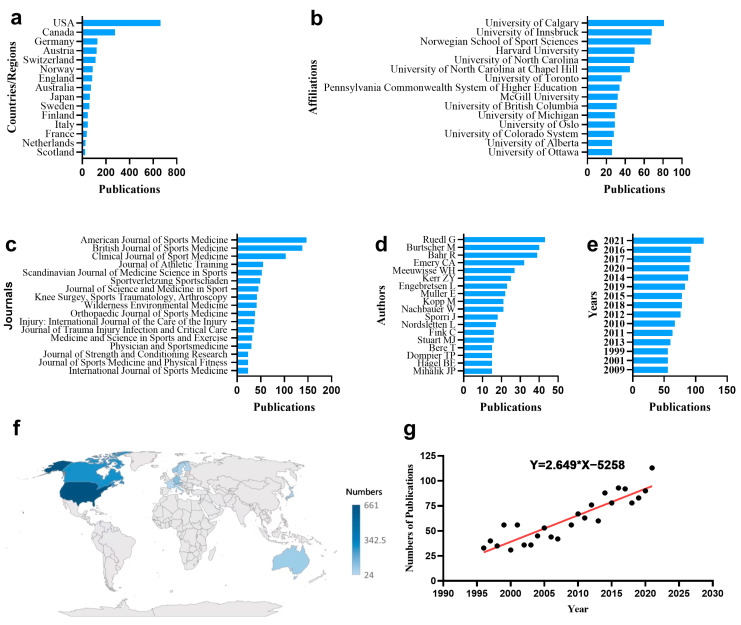
(**a**–**e**) The number and trends of global publications. (**f**) Top 15 countries’/regions’ distribution of ice and snow sports injuries research in world map. (**g**) The linear regression model of trends in the number of publications.

**Figure 2 ijerph-20-02880-f002:**
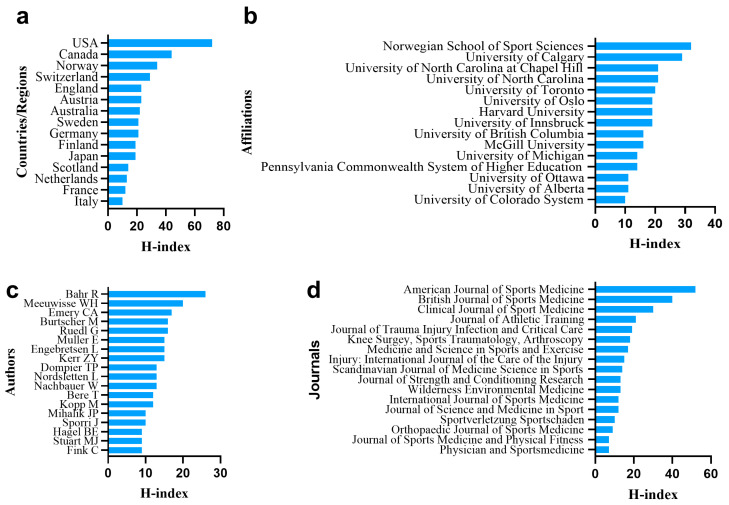
(**a**–**d**) The top 15 countries/regions, affiliations, authors and journals regarding H-index of publications.

**Figure 3 ijerph-20-02880-f003:**
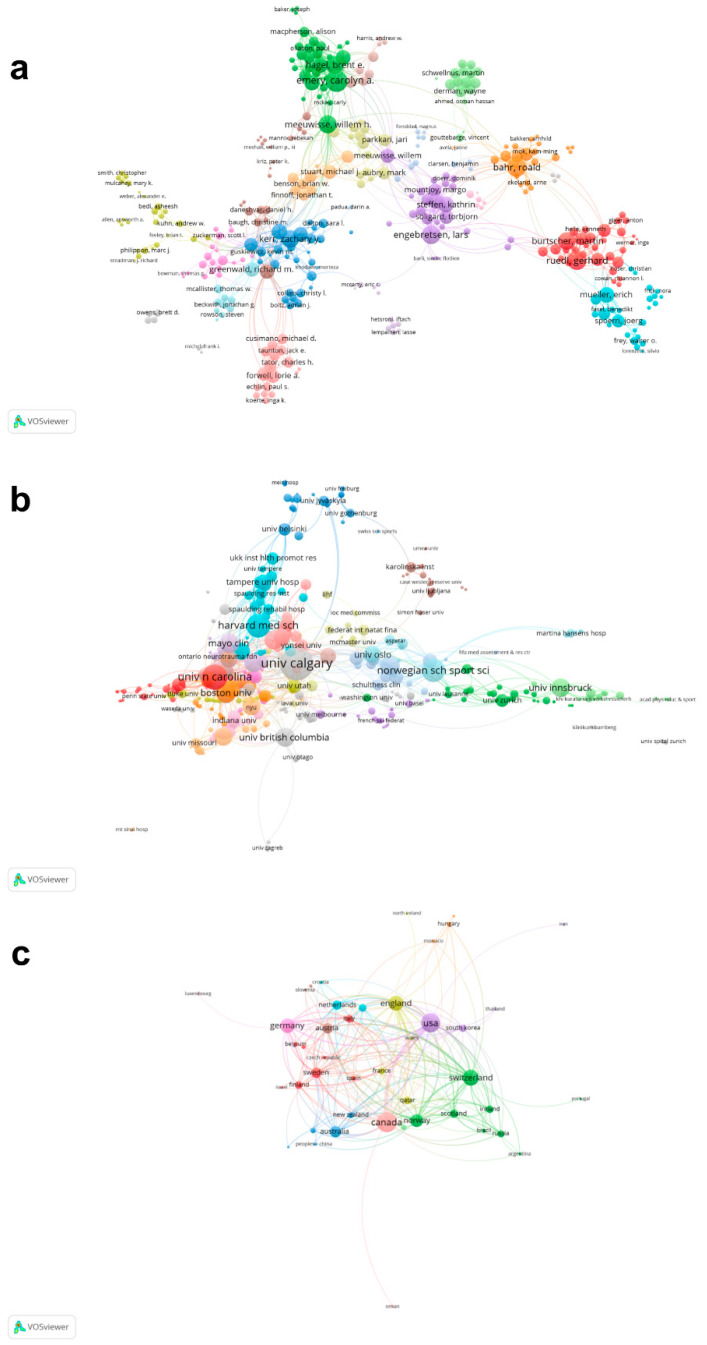
Co-authorship analysis of global research about ice and snow sports injuries. (**a**) Network visualization mapping for the 837 identified authors. (**b**) Network visualization mapping for the 525 identified affiliations. (**c**) Network visualization mapping for the 46 identified countries/regions. The size of the nodes indicates the frequency of co-authorship. Lines between two nodes indicate that those two authors/affiliations/countries/regions have established collaboration ties. The color of an element indicates the cluster it belongs to, and different clusters are indicated by different colors.

**Figure 4 ijerph-20-02880-f004:**
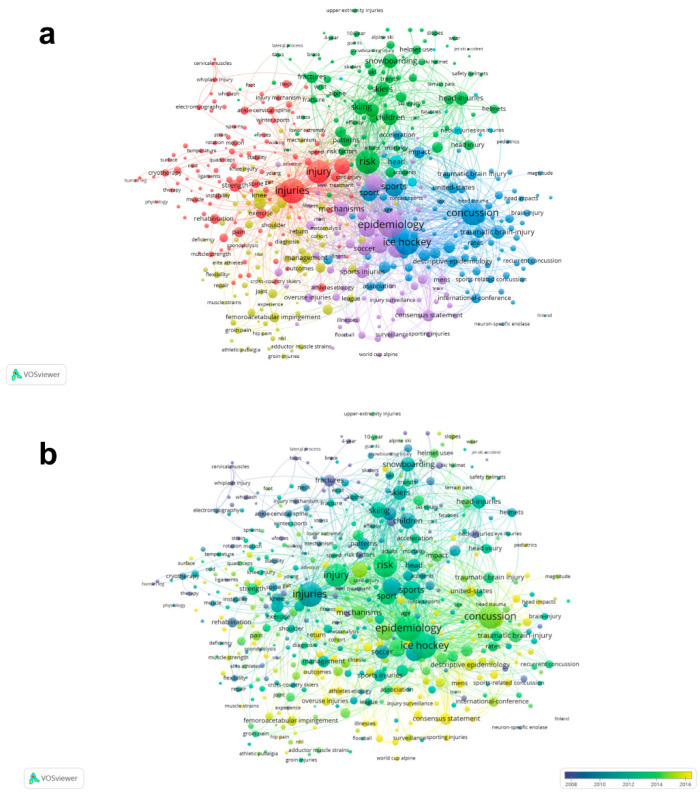
Co-occurrence analysis of global research about ice and snow sports injuries. (**a**) Network visualization mapping for the 233 identified items. Five main clusters were identified: injuries study (red color), head and neck damage study (blue color), risk study (green color), therapy study (yellow color) and epidemiology study (purple color). The size and distance of the nodes indicate the frequency and relationship of items. (**b**) Overlay visualization mapping for the 233 identified items used to show development trends over time.

**Figure 5 ijerph-20-02880-f005:**
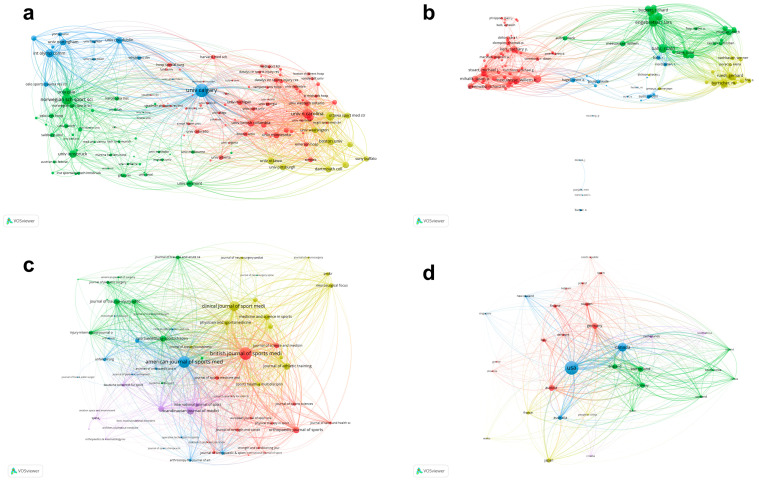
Bibliographic coupling analysis of global research about ice and snow sports injuries. (**a**) Network visualization mapping for the 140 identified affiliations. (**b**) Network visualization mapping for the 128 identified authors. (**c**) Network visualization mapping for the 65 identified journals. (**d**) Network visualization mapping for the 32 identified countries/regions. The size of the nodes and lines between two nodes indicate a proportional relationship to the similarity of the affiliations/authors/journals and countries/regions.

**Figure 6 ijerph-20-02880-f006:**
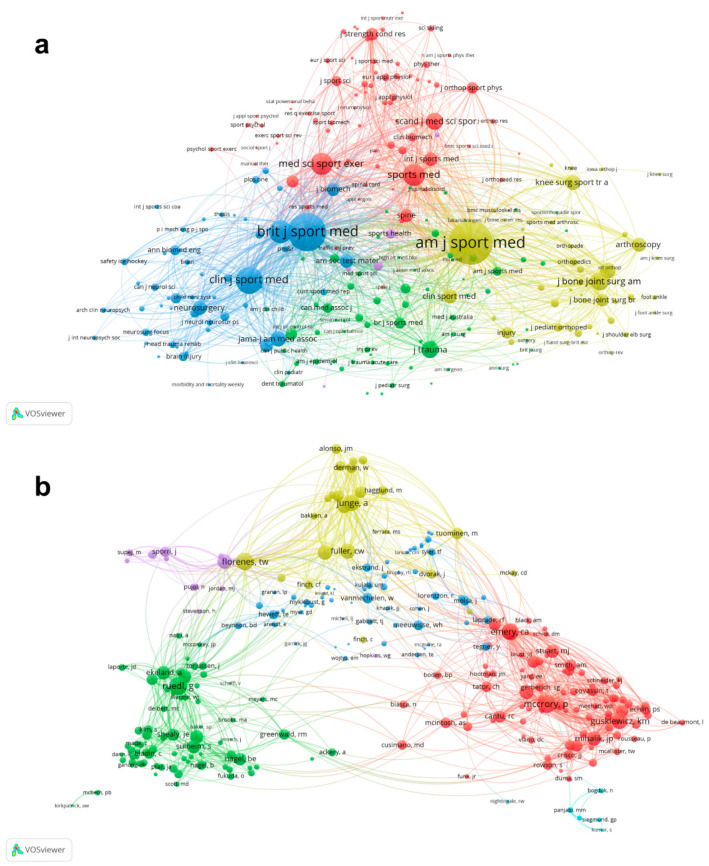
Co-citation analysis of global research about ice and snow sports injuries. (**a**) Network visualization mapping for the 256 identified journals. (**b**) Network visualization mapping for the 260 identified authors. The size of the nodes indicates the frequency of citation. Lines between two nodes indicate that two articles were cited together by other documents.

**Table 1 ijerph-20-02880-t001:** The top 15 most published and cited active countries/regions in the top 3 research directions related to snow and ice sports injuries.

Countries/Regions	Publications (%)	Total Citations	Average Citations Per Item	H-Index
USA	661 (41.184)	20,204	30.57	72
Canada	277 (17.259)	7433	26.83	44
Germany	128 (7.975)	2506	19.58	21
Austria	120 (7.477)	1901	15.84	23
Switzerland	111 (6.916)	3783	34.08	29
Norway	88 (5.483)	3822	43.43	34
England	84 (5.234)	1697	20.2	23
Australia	73 (4.548)	2266	31.04	22
Japan	63 (3.925)	1449	23	19
Sweden	58 (3.614)	1451	25.02	21
Finland	45 (2.804)	1268	28.18	19
Italy	45 (2.804)	393	8.73	10
France	36 (2.243)	557	15.47	12
The Netherlands	27 (1.682)	757	28.04	13
Scotland	24 (1.495)	797	33.21	14

**Table 2 ijerph-20-02880-t002:** The top 15 most published and cited active affiliations in the top 3 research directions related to snow and ice sports injuries.

Affiliations	Number (%)	Country	Total Citations	Average Citations Per Item	H-Index
University of Calgary	81 (5.047)	Canada	3233	39.91	29
University of Innsbruck	68 (4.237)	Austria	1073	15.78	19
Norwegian School of Sport Sciences	67 (4.174)	Norway	3306	49.34	32
Harvard University	50 (3.115)	USA	1657	33.14	19
University of North Carolina	49 (3.053)	USA	1605	32.76	21
University of North Carolina at Chapel Hill	45 (2.804)	USA	1391	30.91	21
University of Toronto	36 (2.243)	Canada	825	22.92	20
Pennsylvania Commonwealth System of Higher Education	34 (2.118)	USA	1752	51.53	14
McGill University	32 (1.994)	Canada	1596	49.88	16
University of British Columbia	31 (1.931)	Canada	936	30.19	16
University of Michigan	29 (1.807)	USA	478	16.48	14
University of Oslo	29 (1.807)	Norway	1480	51.03	19
University of Colorado System	28 (1.745)	USA	1130	40.36	10
University of Alberta	26 (1.62)	Canada	444	17.08	11
University of Ottawa	26 (1.62)	Canada	320	12.31	11

**Table 3 ijerph-20-02880-t003:** The top 15 most published and cited productive authors in the top 3 research directions related to snow and ice sports injuries.

Authors	Publications (%)	Total Citations	Average Citations Per Item	H-Index	Affiliation
Ruedl G	43 (2.679)	756	17.58	16	University of Innsbruck
Burtscher M	40 (2.492)	746	18.65	16	University of Innsbruck
Bahr R	39 (2.43)	2537	65.05	26	Norwegian School of Sport Sciences
Emery CA	32 (1.994)	984	30.75	17	University of Calgary
Meeuwisse WH	27 (1.682)	1369	50.7	20	University of Calgary
Kerr ZY	25 (1.558)	928	37.12	15	University of North Carolina
Engebretsen L	23 (1.433)	1484	64.52	15	Norwegian School of Sport Sciences
Muller E	22 (1.371)	542	24.64	15	Salzburg University
Kopp M	21 (1.308)	368	17.52	12	University of Innsbruck
Nachbauer W	21 (1.308)	473	22.52	13	University of Innsbruck
Sporri J	18 (1.121)	332	18.44	10	Salzburg University
Nordsletten L	17 (1.059)	676	39.76	13	Norwegian School of Sport Sciences
Fink C	16 (0.997)	269	16.81	9	University of Innsbruck
Stuart MJ	16 (0.997)	404	25.25	9	Mayo Clinic
Bere T	15 (0.935)	555	37	12	Norwegian School of Sport Sciences
Dompier TP	15 (0.935)	600	40	13	University of North Carolina
Hagel BE	15 (0.935)	236	15.73	9	University of Calgary
Mihalik JP	15 (0.935)	302	20.13	10	University of North Carolina

**Table 4 ijerph-20-02880-t004:** The top 15 most published and cited productive journals in the top 3 research directions related to snow and ice sports injuries.

Publication Journal	Number (%)	Country	IF (2021)	JCR (2021)	Total Citations	Average Citations Per Item	H-Index
American Journal of Sports Medicine	147 (9.159)	USA	6.202	Q1	8584	58.39	52
British Journal of Sports Medicine	138 (8.598)	England	13.800	Q1	5728	41.51	40
Clinical Journal of Sport Medicine	103 (6.417)	USA	3.638	Q1	2938	28.52	30
Journal of Athletic Training	55 (3.427)	USA	2.860	Q2	1334	24.25	21
Scandinavian Journal of Medicine Science in Sports	52 (3.24)	Denmark	4.221	Q1	1609	30.94	14
Sportverletzung Sportschaden	49 (3.053)	Germany	1.077	Q4	324	6.61	10
Journal of Science and Medicine in Sport	45 (2.804)	Austrilia	4.319	Q1	467	10.38	12
Knee Surgery, Sports Traumatology, Arthroscopy	42 (2.617)	Germany	4.342	Q1	1071	25.5	18
Wilderness Environmental Medicine	41 (2.555)	USA	1.518	Q3	506	12.34	13
Orthopaedic Journal of Sports Medicine	38 (2.368)	USA	2.727	Q2	233	6.13	9
Injury: International Journal of the Care of the Injured	37 (2.305)	England	2.586	Q4	616	16.65	15
Journal of Trauma Injury Infection and Critical Care	35 (2.181)	USA	NA	NA	1088	31.09	19
Medicine and Science in Sports and Exercise	32 (1.994)	USA	5.411	Q1	1055	32.97	17
Physician and Sportsmedicine	30 (1.869)	USA	2.241	Q3	168	5.6	7
International Journal of Sports Medicine	23 (1.433)	Germany	3.118	Q2	600	26.09	12
Journal of Sports Medicine and Physical Fitness	23 (1.433)	Italy	1.637	Q4	160	6.96	7
Journal of Strength and Conditioning Research	23 (1.433)	USA	3.775	Q1	435	18.91	13

## Data Availability

The datasets generated and analyzed during the present study are available from the corresponding author on reasonable request.

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
