# Peer review of "The Global Research Status and Trends in Ice and Snow Sports Injuries from 1995 to 2022: A Bibliometric and Visualized Analysis"

_ijerph, 2023, doi:10.3390/ijerph20042880_

Round 1

Reviewer 1 Report

This submission needs significant linguistic corrections. In my review, I will point out some (just some)  of them.

The scientific method used by the authors is adequate but standard.

The article is largely written in the past tense, which would – wrongly – mean that what is observed was valid in the past but not anymore now. This is an important linguistic issue. An example is on line 37 where the authors write that snow sports have been popular worldwide.  This means that snow sports are not popular anymore.

Line 38. The word “it’’  refers to persons practicing ice and snow sports. Its use is grammatically incorrect.

It is “a bibliometric study” not a “ bibliometrics study”

Line 63  Which Core Collection?  Available data may depend on what the university pays for.  Starting dates may change (some universities have access to Core Collection data starting in 1900 and for others, this starting date is much more recent.  Many Chinese universities do not have access to Proceedings or to the Book Citation Index, although they belong to the Core Collection. Please be precise.

Line 78. The definition of the h-index as given by the authors is not correct.

If an h-index for journals is given, does this number refer to all articles in these journals, or only to the set of articles studied by the authors?

Rankings of universities. The University of North Carolina is a system of Universities (like the University of California) among which the one at Chapel Hill is the most important. You cannot have The University of North Carolina and the University of North Carolina at Chapel Hill in the same ranking as these two entities overlap.

Line 250. Bibliometrics (not ‘the’ bibliometrics) was not introduced in 1969, but the term ‘bibliometrics’ was introduced in that year. And that too is not correct as the term had been introduced much earlier, see (Rousseau, Nature, 510(7504), 12 June 2014, p.218 ) but Pritchard was not aware of this.

Discussion. It seems logical that ice and snow sports are performed and hence its consequences, such as injuries, are studied in habitable places where there is snow and ice. This automatically leads to Northern America and Northern Europe and mountainous regions (such as Switzerland).  It is logical that African, Latin American and  Arabic  countries are less interested in snow and ice sports. This also holds for India and Australia. Hence, this observation (conclusion) is not special.

Line 298  How do the authors know that some influential articles were cited a limited number of times?  What has Garfield to do with their specific study?

Line 375 The article by Kessler has been published in the journal “American Documentation”

Author Response

Thank you for offering us an opportunity to improve the quality of our submitted manuscript (ID: ijerph-2085088, The Global Research Status and Trends in the Ice and Snow Sports Injuries from 1995 to 2022: A Bibliometric and Visualized Analysis). We appreciated very much the reviewers’ constructive and insightful comments.

We have studied comments carefully and have made correction which we hope meet the approval. We highlighted all the revisions in blue in the revised manuscript. In the following, our point-to-point responses to the queries raised by the reviewers are listed in the Responses to reviewers.

Thank you and best regards,

Your sincerely,

Reviewer 2 Report

Overall evaluation

This study evaluates the global research status and trends in ice and snow sports injuries from 1995 to 2022 using a bibliometric and visualized analysis of winter sports-related injuries. This work offers a very useful basis for the understanding of the state and trends of the research with relevant detailed information on the ice and snow sports injuries related literature. This is a generally well-conducted study that provides interesting findings based on a large panel of articles and on a long period of time. Nevertheless, some points of the methods need to be clarified and the discussion of the results does not always stay in the scope of the data.

Major points

1.     Lines 54-59, lines 247-248: The present work does not allow to understand the incidence, characteristics, mechanisms, and risk factors of winter sport injuries. This can only be achieved through a systematic review or a meta-analysis of the literature. Please rephrase or remove.

2.     Line 70-73: The selection of the 1605 articles should be clarified with clear criteria. How many articles were excluded and based on which criteria? This information should be supported with a flow diagram (see for example the PRISMA flow diagrams)

3.     Lines 181-183: The method for the identification of the 5 clusters should be detailed, as well as the process of cluster naming/labeling. Furthermore, the keywords in each cluster appear diverse. This aspect should be discussed.

Minor points

1.     Lines 42-47: Please clarify the relation between the aim of the present study and the Olympic injury and illness surveillance system.

2.     Lines 311-313: This is out of the scope of this study. Please rephrase or remove.

Author Response

(The authors gave the same response as above.)

Round 2

Reviewer 1 Report

The authors performed required changes. In this form it is acceptable, in my opinion. 

Author Response

Thanks very much for your kind work and consideration on publication of our paper. On behalf of my co-authors, we would like to express our great appreciation to reviewers.

Reviewer 2 Report

I thank the authors for the revised version of the manuscript. Although the manuscript has been improved, there are still a few points to clarify.

Major points

1.     Please clarify the selection of the language of publication. In the abstract, it is stated that “articles published in English between 1995 and 2022 were selected”; in the methods section, language of publication is not specified as a selection criterion; in the flow chart, it is specified that articles not written in English were excluded; in the results section, it is written that 89 articles are written in German (lines 127-128).

2.     Lines 57-60, lines 247-248: the authors mean that based on the articles selected in the present study, it would be possible to analyze the evolution of the incidence, characteristics, mechanisms, and risk factors of winter sport injuries. However, this would not be methodologically sound. As mentioned in the first review of this paper, this can only be achieved through a systematic review or a meta-analysis of the literature. This cannot be the aim of a bibliometric analysis.

Minor point

1.     Would it be possible to increase the resolution and/or the size of the figures? Some words are not readable at all.

Author Response

Thank you for offering us an opportunity to improve the quality of our submitted manuscript (ID: ijerph-2085088, The Global Research Status and Trends in the Ice and Snow Sports Injuries from 1995 to 2022: A Bibliometric and Visualized Analysis). We appreciated very much the reviewers’ constructive and insightful comments.

Round 3

Reviewer 2 Report

Minor revision: Please add in the method section that articles published in English were selected for this study.

Author Response

Thank you for offering us an opportunity to improve the quality of our submitted manuscript (ID: ijerph-2085088, The Global Research Status and Trends in the Ice and Snow Sports Injuries from 1995 to 2022: A Bibliometric and Visualized Analysis). We appreciated very much the reviewers’ constructive and insightful comments.

Responses to reviewers (original comments by reviewers are in blue color)

We have studied comments carefully and have made correction which we hope meet the approval. We highlighted all the revisions in blue in the revised manuscript.

Reviewer # 2:

Minor revision: Please add in the method section that articles published in English were selected for this study.

Response: Thanks for your suggestion. We have added “articles published in English were selected for this study” in the method section according to your requirements in the revised manuscript. And we hope the revised manuscript could be acceptable to meet the approval.